# Pharmacists’ Knowledge and Intention to Provide Palliative Care Services in Saudi Arabia: Using the Theory of Planned Behaviour

**DOI:** 10.3390/healthcare11152173

**Published:** 2023-07-31

**Authors:** Ahmed M. Alshehri, Yasser S. Almogbel, Rana E. Alonazi, Waleed M. Alshehri, Hind A. Alkhelaifi, Salman A. Almutairi, Omar S. Alenazi, Ali Z. Alali

**Affiliations:** 1Clinical Pharmacy Department, College of Pharmacy, Prince Sattam bin Abdulaziz University, Al-Kharj 16273, Saudi Arabia; smalmutairi@kfshrc.edu.sa (S.A.A.); aoalanazi@kfshrc.edu.sa (O.S.A.); a.alali@lemonpharmacy.com (A.Z.A.); 2Department of Pharmacy Practice, College of Pharmacy, Qassim University, Buraidah 51452, Saudi Arabia; y.almogbel@qu.edu.sa; 3Pharmacology Department, College of Pharmacy, Prince Sattam bin Abdulaziz University, Al-Kharj 16273, Saudi Arabia; re.alonazi@psau.edu.sa; 4Clinical Pharmacy Department, King Fahad Medical City, Riyadh 11525, Saudi Arabia; walshehri@kfmc.med.sa; 5Contracts Management Department, National Unified Procurement Company (NUPCO), Riyadh 12251, Saudi Arabia; hakhelaifi@nupco.com

**Keywords:** palliative care, pharmacist intention, end of life, healthcare provider’s intention, life-threatening illness, pain management

## Abstract

Providing palliative care to patients with life-threatening illnesses requires multidisciplinary efforts from different healthcare providers. Identifying the attitude, knowledge, and intentions of pharmacists to provide this service in Saudi Arabia is essential. Therefore, this study aimed to identify the palliative care knowledge, intentions, attitudes, subjective norms, and perceived behavioural control of pharmacists and what factors predict their intentions. Cross-sectional questionnaires based on the theory of planned behaviour were distributed to pharmacists in hospitals and community pharmacies. They included items that measured palliative care knowledge, attitudes, intentions, subjective norms, and the perceived behavioural control of pharmacists and identified other sociodemographic and pharmacy-practice-related items. In total, 131 pharmacists completed the questionnaires, showing an average score on palliative knowledge (8.82 ± 1.96; range: 1–14), strong intentions (5.84 ± 1.41; range: 1–7), positive attitudes (6.10 ± 1.47; range: 1–7), positive subjective norms (5.31 ± 1.32; range: 1–7), and positive perceived behavioural control (5.04 ± 1.21; range: 1–7). Having completed a pharmacy residency program, working longer hours per week, having a more positive attitude, and perceived stronger subjective norms were significantly associated with a strong intention to provide palliative care services. Therefore, enabling and motivating pharmacists to complete pharmacy residency programs and improve their attitudes could increase their intentions to provide these services.

## 1. Introduction

End-of-life care includes providing care to individuals whose death is imminent. It varies across cultures or regions and comprises four categories according to the World Health Organization (WHO) standards. The first category encompasses life-threatening conditions for which curative treatment may be feasible but not guaranteed, such as various cancer types and irreversible organ failures of the heart, liver, and kidney. The second category includes conditions in which premature death is predictable but intensive treatment may be used to extend survival and enable participation in daily activities. The third category includes progressive conditions without curative treatment options, where treatment is exclusively palliative and may be necessary for several years. The last category includes irreversible but non-progressive conditions that cause severe disability and increase the risk of health complications and premature death, such as severe cerebral palsy and brain or spinal cord injury [1,2]. Providing care for such individuals is essential and considered a part of palliative care.

In 1990, the WHO recognised palliative care as a unique speciality committed to improving the quality of life of patients with major injuries or life-threatening diseases. Its goals include assessing, preventing, and treating psychological, spiritual, and physical problems. Furthermore, palliative care is distinct from hospice care [3]. Patients receiving palliative care are at a high risk of experiencing adverse events, such as medication misadventures, which may lead to hospitalisation [4,5]. Consequently, the number of hospitals offering palliative care has increased to meet the demand for healthcare. In the US, 25% and 75% of large hospitals provided palliative care services in 2000 and 2015, respectively [3]. The increasing number of hospitals providing palliative care indicates the need for more healthcare providers trained in this field [6,7]. In addition, because palliative care is an evolving speciality, it is in high demand from all healthcare providers [8,9]. Pharmacist expertise in managing medications and preventing medication errors makes them suitable palliative care provider candidates [10,11,12].

Pharmacists are now providing pharmaceutical services, including medication management and psychosocial support to patients, beyond their traditional services [10,11,12,13,14,15,16]. These palliative care services include monitoring adverse events and drug regimens (especially for opioid drugs), improving drug adherence, and providing educational services to cancer patients [3,6,7,14,17].

However, in Saudi Arabia, there is a shortage of evidence-based research regarding pharmacists’ intention to provide palliative care and the factors that affect their intentions. Previous studies have demonstrated that inadequate knowledge, lack of training skills, and attitude toward providing palliative care may limit the contributions of pharmacists in this area [3,18,19]. Therefore, this study aimed to comprehensively evaluate the intentions of hospital pharmacists to provide palliative care in Saudi Arabia and explore factors that may deter their involvement.

## 2. Materials and Methods

### 2.1. Research Setting

This cross-sectional study identified pharmacist intentions and the factors that affect their provision of palliative care. The study included pharmacists practising in hospitals and community pharmacies in Saudi Arabia and was approved by the Saudi Arabian Ministry of Health Central Institutional Review Board (IRB) (No: 20-162E).

### 2.2. Data Collection

The process of data collection involved two methods. First, the central IRB of the Ministry of Health sent the survey to hospital pharmacies and asked them to encourage their pharmacists to complete it. Second, the study authors visited hospitals and community pharmacies to distribute the survey and asked staff pharmacists to complete it online or on hard copies. During the same visits, fliers with a barcode for the study survey were left with the staff pharmacists, who were asked to distribute them to their colleagues.

### 2.3. Survey Development

The survey was developed based on the theory of planned behaviour (TPB) and modified according to the health services research guidelines by Francis et al. [20,21]. It comprised three sections. The first section identified pharmacist intentions, attitudes, subjective norms, and behavioural perceptions towards providing palliative care services; the second section evaluated their knowledge of palliative care [3]; and the third section identified their sociodemographic characteristics and pharmacy-related factors.

The first section included four primary constructs of the TPB. The first construct measured pharmacist intention to provide palliative care, defined as the expected probability degree of providing palliative care. It included three items on a 7-point Likert scale ranging from 1 = strongly disagree to 7 = strongly agree. The average total of these items measured participant intention to provide palliative care. The second construct identified pharmacist attitudes, defined as the degree of positive or negative value placed on providing palliative care to patients with life-threatening illnesses. It included four items on four different 7-point Likert scales (1 = harmful to 7 = beneficial, 1 = bad to 7 = good, 1 = unpleasant to 7 = pleasant, and 1 = worthless to 7 = useful). The average total score of all items identified positive or negative attitudes toward palliative care. The third construct was used for subjective norms, measuring social pressure to provide palliative care. It included four items on 7-point Likert scales; one item ranged from 1 = I should not to 7 = I should, and three ranged from 1 = strongly disagree to 7 = strongly agree. The average total score of these four items measured the perceptions of social pressure to provide palliative care.

The last construct was perceived behavioural control, defined as perceived control over providing palliative care. It included four items on 7-point Likert scales; three items ranged from 1 = strongly disagree to 7 = strongly agree, and one from 1 = easy to 7 = difficult. The average total score of these four items measured the perceived control of the participants over providing palliative care.

The second section included 14 items in a dichotomous (yes/no) format measuring the palliative care knowledge of the participants based on and modified from Adisa et al. [3]. To correctly answer this section, participants had to mark items 2, 6, 7, 10, 12, and 14 as “true” and items 1, 3, 4, 5, 8, 9, 11, and 13 as “false”. The higher the score, the greater the knowledge of the participants about providing palliative care.

The third section identified the sociodemographic characteristics of the participants, pharmacy education, experiences, and settings. The sociodemographic section collected data on sex, marital status, age, current job, geographical region, total monthly income, and citizenship. The pharmacy education section included items that identified education level (BSc., Pharm.D., MSc, or PhD), pharmacy residency, residency speciality, and country where their most recent degree was earned. Pharmacy-related experience and setting included items that identified the years of pharmacy experience, average working hours per week, variables related to previous palliative care services, having relatives who had previously received palliative care, and job setting.

The survey was pilot tested on a convenience sample of 10 hospital and community pharmacists to assess the face and content validity of the study. Respondents were asked to complete the questionnaires and provide comments and feedback on the relevance and clarity of the items. Minor modifications were made after the pilot tests to improve the clarity and order of some items. Cronbach’s alpha was used to measure the internal consistency reliability.

### 2.4. Data Analysis

STATA 16 software was used to perform statistical analyses. Descriptive analyses, means, and frequency distributions were used to describe the study variables. Simple linear regression analysis was used to determine the associations between the independent and dependent variables (intention to provide palliative care). The independent variables included attitude, subjective norms, behavioural control, palliative care knowledge, sociodemographic characteristics, pharmacy education, experience, and setting. Only variables with *p*-values < 0.2 were included in the multivariate linear regression analysis to identify all factors associated with providing palliative care [22]. Variables with *p*-values < 0.05 were considered statistically significant.

## 3. Results

### 3.1. Demographic and Practice Setting Characteristics

The survey was completed by 131 participants. Most were male (61.07%), never married (51.91%), and of Saudi nationality (91.60%). Their average age was 31.16 (±7.85) years, and approximately half (49.62%) were in the centre of Saudi Arabia. Most participants (61.83%) had a monthly income between SAR 5001 and 15,000 (USD 1333.3–4000/month).

The participants had different pharmacy education levels and experiences and worked in various pharmacy settings; 45.80% had a bachelor’s degree, and 41.98% had a pharmacy doctorate (Pharm.D.). Furthermore, most participants (88.55%) had earned their most recent degree in Saudi Arabia.

Most participants (80.15%) did not complete a pharmacy residency program; only 16.79% and 3.05% completed a general and specialised pharmacy residency program, respectively. Participants had a median pharmacy year of experience of 3 years (IQR = 1, 8) and they worked 38.03 (±19.14) h per week on average. Almost half of the participants worked at hospital pharmacies (50.38%), among which 50.00% were staff pharmacists; 36.36% worked in inpatient pharmacies, and 56.10% worked in tertiary hospitals. Of the 65 participants who worked in a community pharmacy, 76.92%, 13.85%, and 6.15% were staff pharmacists, pharmacy managers, and pharmacy supervisors, respectively (Table 1).

### 3.2. Theory of Planned Behaviour Constructs

The intentions, attitudes, subjective norms, and behavioural perceptions of participants to provide palliative care services showed a reliability score ranging from 0.64 to 0.88 (Table 2).

#### 3.2.1. Intentions to Provide Palliative Care

Intentions to provide palliative care were high (mean = 5.84 ± 1.41; range: 1–7). More than half of the participants strongly agreed that they expected, wanted, and intended to provide palliative care to patients with life-threatening illnesses.

#### 3.2.2. Attitude toward Providing Palliative Care

Participants showed a positive attitude toward providing palliative care (overall mean = 6.10 ± 1.47; range: 1–7). They believed that providing palliative care was beneficial (mean = 6.15 ± 1.29; range: 1–7), good (mean = 6.15 ± 1.23; range: 1–7), pleasant (mean = 5.99 ± 1.41; range: 1–7), and useful to patients (mean = 6.11 ± 1.29; range: 1–7).

#### 3.2.3. Subjective Norms

Participants perceived encouragement from people who are important to them to provide palliative care (overall mean = 5.31 ± 1.32; range: 1–7). They believed that most people who are important to them thought (mean = 5.60 ± 1.47; range: 1–7), expected (mean = 5.93 ± 1.45; range: 1–7), pressured (mean = 4.39 ± 2.11; range: 1–7), and wanted them to provide palliative care (mean = 5.33 ± 1.78; range: 1–7).

#### 3.2.4. Perceived Behavioural Control over Providing Palliative Care

Participants were confident in their ability to provide palliative care to patients with life-threatening illnesses (overall mean = 5.04 ± 1.21; range: 1–7). Perceived control over providing palliative care in pharmacy settings was above neutral for all items. They agreed that they had good confidence (mean = 5.77 ± 1.52; range: 1–7), slightly agreed that it was easy (mean = 4.76 ± 1.67; range: 1–7), slightly agreed that it was out of their control (mean = 4.91 ± 1.73; range: 1–7), and slightly agreed that it was entirely up to them to provide palliative care (mean = 4.70 ± 2.01; range: 1–7).

**Table 2 healthcare-11-02173-t002:** Descriptive statistics and reliability for the theory of planned behaviour construct (*N* = 131).

Items	Mean (±SD)	Frequency *n* (%)						
		1	2	3	4	5	6	7
Intention								
(a) I expect to provide palliative care for my patients with life-threatening illnesses. ^1^	5.76 (±1.66)	5 (3.82)	3 (2.29)	4 (3.05)	17 (12.98)	19 (14.50)	12 (9.16)	71 (54.20)
(b) I want to provide palliative care for my patients with life-threatening illnesses. ^1^	5.96 (±1.57)	4 (3.05)	3 (2.29)	2 (1.53)	14 (10.69)	19 (14.50)	9 (6.87)	80 (61.07)
(c) I intend to provide palliative care for my patients with life-threatening illnesses. ^1^	5.79 (±1.57)	2 (1.53)	6 (4.58)	3 (2.29)	16 (12.21)	21 (16.03)	14 (10.69)	69 (52.67)
Domain Average Total	5.84 (±1.41)	Cronbach’s alpha = 0.85						
Attitude								
(a) Providing palliative care for patients with life-threatening illnesses is… ^2^	6.15 (±1.29)	2 (1.53)	0 (0)	4 (3.05)	9 (6.87)	18 (13.74)	20 (15.27)	78 (59.54)
(b) Providing palliative care for patients with life-threatening illnesses is… ^3^	6.15 (±1.23)	1 (0.76)	1 (0.76)	2 (1.53)	11 (8.40)	20 (15.27)	20 (15.27)	76 (58.02)
(c) Providing palliative care for patients with life-threatening illnesses is… ^4^	5.99 (±1.41)	2 (1.53)	2 (1.53)	3 (2.29)	14 (10.69)	19 (14.50)	18 (13.74)	73 (55.73)
(d) Providing palliative care for patients with life-threatening illnesses is… ^5^	6.11 (±1.29)	1 (0.76)	1 (0.76)	3 (2.29)	14 (10.69)	17 (12.98)	18 (13.74)	77 (58.78)
Domain Average Total	6.10 (±1.47)	Cronbach’s alpha = 0.88						
Subjective norms								
(a) Most people who are important to me think that -------------- provide palliative care for my patients with life-threatening illnesses. ^6^	5.60 (±1.47)	3 (2.29)	1 (0.76)	4 (3.05)	26 (19.85)	20 (15.27)	26 (19.85)	51 (38.93)
(b) It is expected of me that I will provide palliative care for my patients with life-threatening illnesses. ^1^	5.93 (±1.45)	1 (0.76)	2 (1.53)	9 (6.87)	13 (9.92)	14 (10.69)	21 (16.03)	71 (54.20)
(c) I feel under social pressure to provide palliative care for my patients with life-threatening illnesses. ^1^	4.39 (±2.11)	19 (14.50)	14 (10.69)	10 (7.63)	19 (14.50)	24 (18.32)	17 (9.92)	32 (24.43)
(d) People who are important to me want me to provide palliative care for my patients with life-threatening illnesses. ^1^	5.33 (±1.78)	8 (6.11)	4 (3.05)	6 (4.58)	23 (17.56)	15 (11.45)	28 (21.37)	47 (35.88)
Domain Average Total	5.31 (±1.32)	Cronbach’s alpha = 0.76						
Perceived behaviour control								
(a) I am confident that I can provide palliative care for my patients with life-threatening illnesses if I want to. ^1^	5.77 (±1.52)	3 (2.29)	4 (3.05)	2 (1.53)	17 (12.98)	20 (15.27)	24 (18.32)	61 (46.56)
(b) For me to provide palliative care for my patients with life-threatening illnesses is ----------. ^7^	4.76 (±1.67)	5 (3.82)	10 (7.63)	10 (7.63	32 (24.43)	32 (24.43)	13 (9.92)	9 (22.14)
(c) The decision to provide palliative care for my patients with life-threatening illnesses is beyond my control. ^1^	4.91 (±1.73)	10 (7.63)	2 (1.53)	7 (5.34)	37 (28.24)	20 (15.27)	25 (19.08)	30 (22.90)
(d) Whether I provide palliative care for my patients with life-threatening illnesses is entirely up to me. ^1^	4.70 (±2.01)	15 (11.45)	6 (4.58)	13 (9.92)	26 (19.85)	16 (12.21)	19 (14.50)	36 (27.48)
Domain Average Total	5.04 (±1.21)	Cronbach’s alpha = 0.64						

SD = standard deviation. ^1^ Response scale: 1 = strongly disagree to 7 = strongly agree. ^2^ Response scale: 1 = harmful to 7 = beneficial. ^3^ Response scale: 1 = bad to 7 = good. ^4^ Response scale: 1 = unpleasant for me to 7 = pleasant for me. ^5^ Response scale: 1 = worthless to 7 = useful. ^6^ Response scale: 1 = I should not to 7 = I should. ^7^ Response scale: 1 = difficult to 7 = easy.

#### 3.2.5. Palliative Care Knowledge

Participant palliative care knowledge was slightly above average (8.82 ± 1.96; range: 1–14). More than half of the participants were unaware that palliative care was not limited to patients without curative treatment options (51.91%), to doctors and nurses (78.63%), or to patients near death (71.76%), did not involve only pain management (64.89%), and were concerned about the safety and effectiveness of non-prescription medications of patients (65.65%) (Table 3).

### 3.3. Predictors of Pharmacist Intentions to Provide Palliative Care

The linear regression analysis showed that only certain variables (age, currently working in a certain geographical region (Riyadh vs. not in Riyadh), income, type of pharmacy setting (community vs. non-community pharmacy), pharmacy residency program (completed vs. none), average pharmacy experience, whether a relative had received palliative care, average working hours per week, attitude, subjective norms, perceived behavioural control, and palliative care knowledge) were significantly associated with participant intentions to provide palliative care (*p* < 0.2) (Table 4). However, when these variables were included in the multivariate linear regression, only completing a pharmacy residency program, number of working hours per week, attitude, and subjective norms were significant predictors of participant intentions to provide palliative care when other variables were constant. Completion of a pharmacy residency program positively affected the intention to provide palliative care (β = 0.698; 95% CI, 0.208 to 1.188; *p*-value = 0.006). Participants who worked longer hours per week were more willing to provide palliative care (β = 0.011; 95% CI, 0.001 to 0.022; *p* = 0.038). Participants with a positive attitude were predicted to have a positive intention to provide palliative care (β = 0.418; 95% CI, 0.170 to 0.667; *p* = 0.001). Finally, feeling social pressure predicted intention to provide palliative care (β = 0.336; 95% CI, 0.151 to 0.521; *p <* 0.001) when other variables were constant.

## 4. Discussion

Palliative care helps to optimise the quality of life of patients with severe illnesses and life-threatening conditions and prevent further complications. Pharmacists are well-suited to provide these services [3,6,7,14]. This study found that pharmacists who completed a pharmacy residency program, had a more positive attitude toward palliative care, perceived stronger subjective norms, and worked longer hours per week were more likely to intend to provide palliative care services.

Overall, the participants had positive intentions toward providing palliative care; the average intention to provide palliative care was high (5.84 ± 1.41). Similarly, pharmacists have been shown to positively intend to provide other pharmaceutical services [23,24,25,26]. More than three-quarters of participants agreed that they wanted, expected, and intended to provide palliative care to patients with life-threatening illnesses. Despite the fact that 63.36% reported that they had not previously provided palliative care services, they reported a positive intention to provide this service. Knowing that pharmacists intend to offer pharmaceutical services will help policymakers in the Ministry of Health and hospitals facilitate their provision by eliminating existing barriers.

The attitude of the participants toward providing palliative care was positive; the overall average attitude was high (6.10 ± 1.47). Participants agreed that providing palliative care to patients with life-threatening illnesses was beneficial (6.15 ± 1.29) and good (6.15 ± 1.23). More than 80% of participants agreed that providing palliative care was beneficial, good, useful, and pleasant for patients. Previous studies have found that pharmacists have positive attitudes toward different pharmaceutical services [3,27,28]. For example, a Nigerian study conducted in 2017 showed that pharmacists had a positive attitude toward providing these services [3]. Another in 2022 showed that pharmacists had a positive attitude toward providing diabetic care [27]. Other healthcare providers and patients perceive the palliative care services of pharmacists as beneficial [14].

Individuals important to the participants positively influenced their willingness to provide palliative care; their average subjective norms were above neutral (5.13 ± 1.32). More than 80% of the participants agreed that people important to them thought they should provide palliative care to patients with life-threatening illnesses. Pharmacists were affected by people around them and their thoughts on the importance of pharmaceutical services. Previous studies have demonstrated that subjective norms have the most substantial influence on pharmacist intentions to use adverse event reporting systems and report serious adverse events [23,29]. Providing palliative care involves caring for people who suffer from pain and other unpleasant symptoms and have a low quality of life, which may strengthen the effects of subjective norms.

Participants had a positive perceived behavioural control related to palliative care. More than 80% agreed that they had confidence, and more than 50% agreed that it was easy to provide these services. Pharmacists are generally more willing to be involved in different pharmaceutical services [28]; however, Williams et al. found that pharmacists thought it difficult to report medication safety incidents [30]. More than half of the participants in this study agreed that providing palliative care was beyond their control, which was expected, as 50.77% and 78.13% of hospital and community pharmacists, respectively, worked as staff pharmacists, indicating that they might require a higher authority to implement these services. The American Society of Healthcare Providers has released guidelines on pharmacist rules in palliative and hospice care, which encourage community and hospital pharmacists to participate in their institution communities and proceed to change healthcare practice settings [15].

Palliative care knowledge was slightly above average (8.82 ± 1.96). However, more than half of the participants incorrectly believed that palliative care was limited to patients without curative treatment options or those near death, involved only pain management medications, or was only provided by nurses or physicians. Having limited knowledge of some palliative care knowledge scale items was expected as 63.36% of participants had never offered palliative care services, and 71.76% had no relatives who had previously received or were currently receiving the service. In contrast, more than half of the participants correctly believed that palliative care services aimed to relieve patient symptoms and help them get a good night’s sleep, possibly because they speculated that this service was for terminally ill patients and could be used to ease their lives and relieve symptoms. Similarly, another study found that pharmacists reduced patient pain in palliative care clinics by identifying medication problems for managing pain and constipation [31]. In this study, 83.97% of the participants believed that their involvement could decrease the need for medical emergencies, similar to the findings of other studies that have clarified the importance of incorporating pharmacists in delivering healthcare services to lower the potential for emergencies or unnecessary healthcare expenses [12,32].

Furthermore, this study revealed several factors that affected intentions to provide palliative care. Pharmacist attitudes, subjective norms, hours worked per week, and completing a pharmacy residency program were all associated with their intention to provide these services. Policymakers from the Ministry of Health, hospitals, and community pharmacies could modify these factors to increase the involvement of pharmacists in palliative care; improving their attitudes toward providing palliative care would also increase their intention to deliver these services. In other studies, improving pharmacist attitudes toward various pharmaceutical services increased their intentions to provide these services [25,27,30,33]. Improving societal perspectives, including those of healthcare providers, on palliative care would strengthen the impact of subjective norms on pharmacists’ intention to provide palliative care. Studies have shown that improving the support received from pharmacy supervisors enables them to provide various pharmaceutical services [27,30,33,34]. Furthermore, the lack of time hinders pharmacists from delivering various pharmaceutical services [12,35]. Thus, decreasing their working hours will enable them to allocate more time to provide additional pharmaceutical services, such as palliative care. Furthermore, completing the pharmacy residency program would increase the opportunity for pharmacists to learn clinical skills and provide palliative care and other pharmaceutical services [25]. Providing training courses for pharmacists, or even incorporating palliative care services topics in the pharmacy college curriculum, might replicate the impact of completing a residency program and enhance the pharmacist’s intention to provide palliative care services.

This study had several limitations. First, the participants were asked to complete the survey anonymously; therefore, their responses could not be validated. Second, because the survey barcode was distributed to pharmacies, and participants were asked to complete the survey and share the barcode with other pharmacists, measuring the survey response rate was impossible. Lastly, association does not always imply causation; therefore, the factors found to be possibly associated with intentions to provide palliative care may not be the causation for intention.

This study is the first to identify the knowledge and intention of pharmacists to provide palliative care in Saudi Arabia and explore the factors that might affect their choice using a well-established theoretical model and validated measures. Understanding these factors will guide further research to improve their impact and increase pharmacist intentions to provide these services. Identifying pharmacist intentions is not the ultimate goal; researchers need to determine the prevalence of teaching these services in pharmacy schools, offering these services to patients, and the knowledge of healthcare providers about these services. Finally, identifying the current collaboration patterns between pharmacists and other healthcare providers in delivering palliative care is essential.

## 5. Conclusions

This study identified pharmacist intentions to provide palliative care using the theory of planned behaviour. Pharmacist attitudes, subjective norms, working hours, and completion of a residency program predicted their intentions to provide palliative care. Improving the impact of these factors would increase their intentions to provide these services and make their provision more consistent. Healthcare policymakers need to target these factors to enhance pharmacist involvement in palliative care teams.

## Figures and Tables

**Table 1 healthcare-11-02173-t001:** Sociodemographic and pharmacy practice characteristics of participants (*N* = 131).

Characteristics	*n*	%
Age		
Age (years); mean ± SD	31.16 (±7.85)	
Gender		
Male	80	61.07
Female	51	38.93
Marital status		
Married	58	44.27
Divorced	4	3.05
Separated	1	0.76
Widowed	0	0.0
Never married	68	51.91
Pharmacy job location in the Saudi region		
Riyadh	65	49.62
Makkah	22	16.79
Al-Qassim	15	11.45
Eastern Region	10	7.63
Jazan	4	3.05
The North Border region	4	3.05
Al-Madinah	3	2.29
Al-Baha	3	2.29
Other ^1^	5	3.82
Total monthly income (SAR)		
≤5000	23	17.56
5001–10,000	41	31.30
10,001–15,000	40	30.53
15,001–20,000	11	8.40
20,001–25,000	5	3.82
25,001–30,000	5	3.82
≥30,001	6	4.58
Highest pharmacy education level		
Bachelor’s degree	60	45.80
Pharmacy doctorate degree	55	41.98
Master’s degree	13	9.92
Doctorate degree	2	1.53
Other ^2^	1	0.76
Completed a residency training program		
No	105	80.15
Yes, general residency	22	16.79
Yes, specialised residency	4	3.05
Pharmacy Residency Specialty		
Oncology	1	25.00
Infectious disease	1	25.00
Ambulatory care	1	25.00
Other ^3^	1	25.00
Pharmacy years of experience		
The median pharmacy years of experience; median (IQR)	3 (IQR = 1, 8)	
Citizenship		
Saudi Arabia	120	91.60
Egyptian	9	6.87
Others ^4^	2	1.53
Country where obtained the latest pharmacy degree or training		
Saudi Arabia	116	88.55
United Kingdom	10	7.63
United States	4	3.05
Malaysia	1	0.76
Pharmacy job setting		
Community pharmacy	65	49.62
Hospital pharmacy	66	50.38
Community pharmacy job position		
Staff pharmacist	50	76.92
Pharmacy manager	9	13.85
Pharmacy supervisor	4	6.15
Other ^5^	2	3.08
Hospital pharmacy job position		
Staff pharmacist	33	50.00
Pharmacy manager	13	19.70
Clinical pharmacist	9	13.64
Pharmacy supervisor	7	10.60
Pharmacy residents	3	4.55
Other ^6^	1	1.52
Hospital pharmacy setting		
Inpatient pharmacy	24	36.36
Outpatient pharmacy	22	33.33
Clinical pharmacy	13	19.70
Other ^7^	7	10.61
Healthcare institutions level		
Primary healthcare institutions	15	22.73
Secondary healthcare institutions	12	18.18
Tertiary healthcare institutions	37	56.10
Other ^8^	2	3.03
Working hours per week		
Average hours of work per week (mean ± SD)	38.03 (±19.14)	
Previously provided palliative care services		
Yes	48	36.64
No	83	63.36
Number of years providing palliative care services		
Average number of years providing palliative care services (mean ± SD)	3.08 (±3.02)	
Relative received or is currently receiving palliative care services		
Yes	37	28.24
No	94	71.76

^1^ The five participants were from Hail, Tabuk, Asir, Najran, and Al-Jouf. ^2^ One participant did not report a valid response. ^3^ One participant did not report a valid response. ^4^ Two participants: one Jordanian and one Sudanese. ^5^ One participant did not report a valid response and another participant provided an invalid response. ^6^ One participant did not report a valid response. ^7^ Seven participants: two in a drug information centre, two in a pharmacy directory, and one in a medication safety centre, and two participants did not provide valid responses. ^8^ Two participants did not report a valid response.

**Table 3 healthcare-11-02173-t003:** Palliative care knowledge of participants (*N* = 131).

Item	True ^1^	False ^1^
Palliative care involves providing care only to patients without curative treatment options.	68 (51.91)	63 (48.09)
2. Non-medical practitioners are active participants in palliative care.	91 (69.47)	40 (30.53)
3. Palliative care is to be provided by doctors and nurses alone.	103 (78.63)	28 (21.37)
4. Palliative care is required only for patients who are near death.	94 (71.76)	37 (28.24)
5. Palliative care only involves pain management.	85 (64.89)	35 (35.11)
6. Palliative care involves providing patients with relief from their symptoms.	109 (83.21)	22 (16.79)
7. One of the goals of pain management in palliative care is to get a good night’s sleep.	114 (87.02)	17 (12.98)
8. Palliative care does involve maintaining patient medication profiles over time.	33 (25.19)	98 (74.81)
9. Palliative care should be provided in conjunction with curative care at the time of diagnosis of a potentially life-limiting illness.	12 (9.16)	119 (90.84)
10. The goals of palliative care and pharmaceutical care are consistent.	112 (85.50)	19 (14.50)
11. Medication therapy is the cornerstone of all symptom control in palliative care.	37 (28.24)	94 (71.76)
12. Involvement of pharmacists in palliative care activities may decrease the need for medical emergencies.	110 (83.97)	21 (16.03)
13. Pharmacists in palliative care should be less concerned about monitoring non-prescription medication use for safety and effectiveness.	86 (65.65)	45 (34.35)
14. Pharmacists in palliative care communicate with pharmaceutical manufacturers to determine the availability of nonstandard dosage forms.	102 (77.86)	29 (22.14)
Total Average	8.82 (±1.96)	

^1^ Participants received a perfect score if they marked items 2, 6, 7, 10, 12, and 14 as “true” and items 1, 3, 4, 5, 8, 9, 11, and 13 as “false”.

**Table 4 healthcare-11-02173-t004:** Multivariate linear regression analysis of factors associated with average intention to provide palliative care (*N* = 131).

Variable	Beta	95% Confidence Interval		*p*-Value
		Lower	Upper	
Age	0.037	−0.047	0.122	0.388
Current job geographical region (Riyadh vs. other regions)	−0.343	−0.724	0.038	0.077
Total monthly income (≤SAR 15,000 and >SAR 15,000)	−0.359	−1.017	0.300	0.283
Community pharmacy(Community vs. hospitals)	0.182	−0.256	0.620	0.413
Pharmacy residency program (completed vs. none)	0.698	0.208	1.188	0.006 *
Pharmacy experience (years)	−0.043	−0.138	0.052	0.371
Relatives who received palliative care	−0.208	−0.658	0.243	0.363
Hours worked per week	0.011	0.001	0.022	0.038 *
Attitude	0.418	0.170	0.667	0.001 *
Subjective norms	0.336	0.151	0.521	<0.001 *
Perceived behavioural control	−0.039	−2.030	1.364	0.681
Palliative care knowledge	0.453	−2.091	2.997	0.725

** p* value > 0.05.

## Data Availability

The data that support the findings of this study are available from the corresponding author.

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
