# Peer review of "Pharmacists’ Knowledge and Intention to Provide Palliative Care Services in Saudi Arabia: Using the Theory of Planned Behaviour"

_healthcare, 2023, doi:10.3390/healthcare11152173_

Round 1

Reviewer 1 Report

Overall Comment

Thank you for the opportunity to review this study. This is a compelling and well executed study on pharmacist’s intention to engage in palliative care. It makes an important contribution to the literature by examining factors that drive intention to engage in palliative care and is strengthened by using a well-established theoretical model (the theory of planned behavior). I believe this paper could be strengthened further by painting a clearer conclusion using the suggesting below and by taking a more nuanced approach to how findings from this study could be used to improve the provision of palliative care by pharmacists.

Major:

1.     Abstract, Line 32: I’m concerned that the conclusion of the abstract may not be appropriate. Are residencies a requirement to practice pharmacy in Saudi Arabia? Is it reasonable to motivate all pharmacists to complete residencies? I think the authors should consider reframing their conclusion here.

2.     Table 2: I find this table a bit overwhelming and I’m not sure that providing this level of detail is necessary. The authors might consider only reporting the mean, SD along with the median and IQR or the median and min and max. If the authors want to provide the level of detail of the frequencies, perhaps it can be moved to a supplemental table.

3.     The authors state that significance for the simple linear regression analysis was set at p<0.2 (line 143) in the methods section, and then say it was set at p<0.1 (line 225) in the results. What was significance set at and why?

4.     Paragraph that starts at Line 246: I think your findings that pharmacists would like to provide these services is very compelling. I think that by contrasting the percentage of those who do not provide these services to the reported intention to provide these services will help emphasize your point. Something like “despite 83% reporting that they have not previously provided palliative care services, pharmacists report a positive intention to provide this service.”

5.     I think your study really identifies an important gap. People don’t provide these services but they have positive attitudes, perceived behavioral control, subjective norm AND intention. So what are we missing here?  The authors found that attitudes, training and subjective norms drive intention. I think the authors do a good job of suggesting how we can target these things. With regards to residencies, as mentioned before it may not be feasible to encourage everyone to complete a residency.  Perhaps the ministry could consider some type of targeted training program or a continuing education requirement that is implemented by the ministry?  

6.     I don’t believe the final sentence of the conclusion is appropriate there. It doesn’t really relate to any of the results of the study presented here and was not mentioned anywhere else.

Minor:

1.     Abstract, Line 24: This line lists two different levels. Do the authors mean hospitals and community pharmacies or hospital pharmacists and community pharmacists?

2.     Line 70: consider rewording to “on the extent of the intentions of pharmacists to…” to “regarding pharmacists’ intention to…”

3.     The spacing in the tables needs to be corrected regarding percentages in the last row of each variable.

4.     Table 3: What does the bold in the table represent? If this is to represent the correct answer, I would suggest instead writing the answer in italics and parenthesis after each item. Then also amend the Item label at the top of the table to include “(correct answer)”

5.     Line 117: How was the palliative care knowledge scale modified?

6.     Line: 252: reword to eliminating existing barriers

7.     Line 246: I believe you mean intention not behavior

8.     Line 273: I believe you mean positive perceived behavioral control related to palliative care, not perception.

9.     What do the authors mean that the surveys couldn’t be validated? Most surveys are completed anonymously and the authors used validated measures.

10.  I think the use of a well-established theoretical model and validated measures are strengths that should be noted somewhere, perhaps in paragraph starting on line 326

There are some some edits that are needed regarding the use of the English language. Overall, now major issues. 

Author Response

Dear Reviewer,

Thank you for taking the time to provide us with your feedback. We do truly believe your comment has strengthened our manuscript. We have thoroughly reviewed each comment and have created a detailed response to address all the concerns that were raised.

Regards!

Sincerely,

Ahmed M. Alshehri, B.Pharm., M.S., Ph.D

Associate Professor

Clinical Pharmacy Department, College of Pharmacy, Prince Sattam bin Abdulaziz University, Alkarj, Riyadh, Saudi Arabia

Prince Sattam Bin Abdulaziz University-College of Pharmacy

3987 Al Kharj, 16273-6758 Saudi Arabia

0096115886055

Ah.alshehri@psau.edu.sa

Reviewer 2 Report

Dear Authors,

Palliative care is an extremely important part of medicine and it is becoming more and more important over the time.A broader understanding of this discipline is necessary to meet the health needs of modern society. That is why I consider the research topic you have undertaken to be very important.

Reading the text of your manuscript, I have some doubts that I would like to ask for clarification.

1.     In the summary on lines 27-30 you give the results in absolute numbers. Unfortunately, the reader does not know the reference point/scale you used. Therefore, it is not able to form an opinion whether the presented results are favorable or not.

2.     I noticed a similar ambiguity in line 209. It may be better to give results in percentages, since the scales of individual categories of results are different

3.     In line 149 you indicate the average age of the pharmacologist involved in the study 31.16 ± 7.85 someone was 23.31 years old. Isn't that too little work experience in the profession? This is even a low age for graduating from high school. Does this age affect the objectivity of the study?

4.     Line 151 - earnings in 1 group differ three times (which can significantly change the employee's motivation) and do not refer to the average earnings in the country, the reader from outside Saudi Arabia does not know whether it is high earnings.

5.     Line 159 - average experience in the profession 5.69 ± 7.31 suggests that someone could have had negative experience in the years, which is not possible.

6.     Lines 161-162 - percent add up to more than 100%?

7.     Table 1 - martial status adds up to 102%?

8.     Check the results and their summation carefully, because some of them are not possible.

9.     How would you interpret your result that pharmacists working more hours per week are more motivated to provide palliative care - line 302?

10.  Conclusions is a general summary and no actual conclusions based on the results of own research - a chapter for a thorough reconstruction

Best regards,

Author Response

(The authors gave the same response as above.)

Reviewer 3 Report

The overall manuscript is suggesting the importance of palliative care and its understanding in Saudi Arabia. However, a few suggestions is mentioned below:

1. Abstract needs to be improved especially the background of the study.

2. the results of the study can be presented in a better way like graphs and other possibilities.

3. The most important what next about the study as it covers only a small population. Does it going to change the behavior of pharmacists or it will be shared with authorities to ensure such casualties?

4. Is the questionnaire also reviewed by some authorities who decided these questions is enough for this study?

5. Questionnaire needs to be shared with this article.

English is fine just need few spelling errors whic need to be corrected.

Author Response

(The authors gave the same response as above.)

Reviewer 4 Report

The manuscript has a good sequence and coherence in methodology, results and discussion, an important issue that could be addressed in the discussion is regarding the regulation on the regulation for the management of drugs that can be used illicitly or illegally.

It is a manuscript that analyzes knowledge and attitudes about the possibility that pharmacists perform palliative care therapies. The instrument/survey analyzes attitudinal components, knowledge, and sociodemographic variables under the theory of planned behavior approach and uses linear regressions to search for associations between the variables studied.

The instrument/survey used is based on the theory of planned behavior, which avoids bias due to predisposition during the execution of the survey, which is an important methodological strength. An important weakness, which manifested in the MDPI system was the topic of management of drugs that can be used illicitly or illegally, the possibility to prescribe opioid or narcotic medications for pain management, should always be a regulated issue and performed in the hospital environment.

The last point would be my main point which should be worked on in the discussion.

It is a good structured manuscript

Author Response

(The authors gave the same response as above.)

Round 2

Reviewer 2 Report

Dear Authors,

Thank You for Your kind answers. Unfortunately not all my doubts have been clarified.

Comment 1 

Some of the results are still misleading. The line shows the mean palliative knowledge score of 8.82 +/- 1.96 on average from 1-7, which seems to exceed the maximum possible score.

Comment 2 

You refer in your answer to lines 141-159 and 352-373 of the manuscript, which do not deal with this issue.

Comment 3

Thank you for your explanations. I respect the opinions of young people, but I would be very careful to make any conclusions from forms filled out by someone who has been in the discipline for only a few months. It can be concluded that the system in Saudi Arabia in palliative care is extremely risky to hire a person just finishing school, nothing more.

Comment 4

I respect the numbers, but I still believe that a threefold difference in monthly pay (you noticed) can make a big difference in the motivation of this group of workers, which has an impact on the results of their forms.

Comment 5

I still don't understand how the paticipant can have 5.69 ± 7.31 years of experience (line 159)? Are you still implying, that some participants have almost “minus” two years of experience in this discipline?

Comment 6 

It is a bit disappointing that your research forms (being the only research tool) are not correctly filled in by some participants and completed. This may affect the results.

Comment 7

Thanks for Your update.

Comment 8

It is a bit disappointing that your research forms (being the only research tool) are not correctly filled in by some participants and completed. This may affect the results.

Comment 9

Thank you for your interpretation. What do you think about the suggestion that people who are highly financially motivated (and work and earn significantly more than others, as your research has shown - up to three times) are simply more motivated?

Comment 10

It is not my role to summarize your research, I afraid.  My comment is that the conclusions still are not supported by the research results, but seem to be a general summary of the research.

Concluding my review, I am very sorry that I did not endorse your manuscript for publication, as in my opinion, it still has methodological flaws and misleading for the reader and sometimes impossible data. The conclusions are not supported by the methodology and research results.

Best regards

Author Response

Dear Reviewer,

Thank you for taking the time to provide us with your feedback. We do truly believe your honest comments have strengthened our manuscript. We have thoroughly reviewed each comment and created a detailed response to address all the raised concerns.

Regards!

Sincerely,

Ahmed M. Alshehri, B.Pharm., M.S., Ph.D

Associate Professor

Clinical Pharmacy Department, College of Pharmacy, Prince Sattam bin Abdulaziz University, Alkarj, Riyadh, Saudi Arabia

Prince Sattam Bin Abdulaziz University-College of Pharmacy

3987 Al Kharj, 16273-6758 Saudi Arabia

0096115886055

Ah.alshehri@psau.edu.sa
